# Analytical Performance of a Novel Latex Turbidimetric Immunoassay, “Nanopia TARC”, for TARC/CCL17 Measurement: A Retrospective Observational Study

**DOI:** 10.3390/diagnostics13182935

**Published:** 2023-09-13

**Authors:** Keita Yamashita, Shiori Takebayashi, Wataru Murata, Nao Hirai, Yui Ito, Mayuka Mitsui, Mina Saito, Kei Sato, Miyuki Terada, Noriyasu Niizeki, Akira Suzuki, Kenya Ogitani, Toshihiko Fujikawa, Marie Komori, Nozomi Inoue, Norimitsu Arai, Masato Maekawa

**Affiliations:** 1Department of Laboratory Medicine, Hamamatsu University School of Medicine, Hamamatsu 431-3192, Japan; tshiori@hama-med.ac.jp (S.T.); w.murata@hama-med.ac.jp (W.M.); hirai-n@hama-med.ac.jp (N.H.); itoyui@hama-med.ac.jp (Y.I.); mayuka@hama-med.ac.jp (M.M.); mina.s@hama-med.ac.jp (M.S.); keis2@hama-med.ac.jp (K.S.); tmyk17@hama-med.ac.jp (M.T.); niizeki@hama-med.ac.jp (N.N.); suzukia@hama-med.ac.jp (A.S.); k.ogi@hama-med.ac.jp (K.O.); 2Diagnostic Products Development, Department Research & Development, Sekisui Medical Co., Ltd., Tokyo 103-0027, Japan; fujikawa@sekisui.com (T.F.); marie.komori@sekisui.com (M.K.); nozomi.inoue@sekisui.com (N.I.); n.arai@sekisui.com (N.A.)

**Keywords:** TARC, CCL17, latex turbidimetric immunoassay, atopic dermatitis, biomarker

## Abstract

Thymus- and activation-regulated chemokine (TARC, also known as CCL17) is used as a biomarker for atopic dermatitis. The methods currently used for its measurement are complex, time-consuming, and require large machinery, warranting the need for a method that is simple, has a quick turnaround time, and requires less complex machinery. We evaluated the analytical performance of a novel latex turbidimetric immunoassay method, “Nanopia TARC”, on 174 residual serum samples from patients with skin or allergic diseases. This evaluation included the assessment of the limit of blank/detection/quantification (LOB/D/Q), precision, accuracy, linearity, interference, and commutability between Nanopia TARC and “HISCL TARC”, based on the chemiluminescent enzyme immunoassay (CLEIA) method. The LOB/D/Q values were 13, 57, and 141 pg/mL, respectively. The coefficient of variation of the repeatability was 0.9–3.8%, and that of the intermediate precision was 2.1–5.4%. The total error of the accuracy was 1.9–13.4%. The linearity was 141 and 19,804 pg/mL for TARC. The correlation coefficient between Nanopia TARC and HISCL TARC determined using the Passing–Bablok regression analysis was 0.999. Furthermore, the concordance of diagnostic criteria with AD was 92%. Nanopia TARC was confirmed to have the same analytical performance for TARC measurement as the existing CLEIA method.

## 1. Introduction

Thymus- and activation-regulated chemokine (TARC, also known as CCL17) was discovered in 1996 [1]. It induces leukocyte migration and functions as a ligand for the chemokine receptor, CCR4, expressed in type 2 helper T lymphocytes (Th2) [2,3]. TARC is also a potential biomarker for eosinophilic disorders, such as allergies, autoimmune diseases, angioimmunoblastic T-cell lymphoma, mycosis fungoides, and *Sézary* syndrome [3,4,5,6,7,8,9]. Kakimura et al. [10] revealed for the first time that the serum TARC levels in patients with atopic dermatitis (AD) were substantially higher than those in healthy individuals and were associated with disease activity. Furthermore, TARC was highly expressed in vascular endothelial cells, T cells, and dendritic cells in the dermis of those patients. Subsequent studies showed similar trends, highlighting the importance of TARC as a biomarker for AD [11,12,13,14,15,16]. In 2008, the diagnosis of AD and evaluation of the disease activity using TARC was covered by health insurance in Japan, and to date, TARC measurement is a routine clinical laboratory test used in diagnosis.

AD is caused by a weakened skin barrier function. It is defined as eczema and prurigo caused due to inflammation induced by cytokines and Th2-cell infiltration as a result of TARC production. Patients with AD often have the following atopic predisposition: (1) family history or medical history of allergies, such as allergic rhinitis or asthma and (2) overproduction of IgE antibodies (such as in food allergies) [17]. The pathogenesis of AD is characterized by diverse symptoms and phenotypes, with multiple factors involved in a complex and nonhierarchical manner. *IL1RL1-IL18R1-IL18RAP*, the major histocompatibility complex region, *OR10A3-NLRP10*, *GLB1*, *CCDC80*, *CARD11*, *ZNF365*, and *CYP24A1-PFDN4* were identified as genetic factors for AD in a genome-wide association study in a Japanese population [18]. Furthermore, environmental factors, such as daily exposure to antigens and irritants, lifestyle, humidity, temperature, wool fibers, and stress, are intricately involved in the onset and exacerbation of AD [19]. Appropriate diagnosis, treatment, and monitoring have a significant impact on the quality of life of patients with AD.

A global epidemiological status of the prevalence of AD was reported in the International Study of Asthma and Allergies in Childhood, wherein AD was shown to be highly prevalent from the childhood to adolescence and regional differences in prevalence were highlighted (the prevalence ranged from 1.1% to 18.4% at ages 6–7 years) [20]. Recent studies have reported that the number of patients with AD below the age of 5 years in Eastern Europe is increasing and that the prevalence is high (approximately 15%) in Estonia and Russia [21]. On the contrary, in Japan, the prevalence rate exceeds 10% in children and is also high in young adults in their 20s, indicating that it is a common skin disease [17]. Conventionally, IgE has been used as a clinical biomarker for AD. However, it lacks specificity because it does not reflect short-term changes in the disease status, and having an atopic predisposition does not necessarily indicate progression to AD.

In view of the findings described above, guidelines established by the Japanese Dermatological Association (JDA) recommend the TARC test for diagnosing and monitoring AD [17]. In contrast, the American Academy of Dermatology does not recommend TARC as a biomarker for AD [22]. Furthermore, evidence for TARC as an AD biomarker is poor in both guidelines [17,22]. This is due to the small sample sizes in many studies and the limited knowledge of differential serum TARC levels between AD and other eczema or other atopic conditions [22].

The enzyme-linked immunosorbent assay (ELISA) is the main method used worldwide for TARC quantification. However, this method takes more than 3 h and is complicated, because of which, it is difficult to use in general medical practice and large-cohort studies. In 2014, a measurement system based on the chemiluminescent enzyme immunoassay (CLEIA) using a dedicated automated analyzer (HISCL^®^-5000/2000i) was established to overcome the issues associated with ELISA. However, the equipment required for the CLEIA method is large and may not be usable in clinics or small hospitals. Although the HISCL-automated immunoassay system has been widely introduced in Japan and China, it is not yet popular in Europe and the United States of America. In this study, we evaluated the analytical performance of a rapid (requiring 10 min) and simple system for TARC measurement developed using a latex turbidimetric immunoassay (LTIA) that can be installed in a general clinical chemistry analyzer.

## 2. Materials and Methods

### 2.1. Subject Specimens and Study Design

We used 174 residual serum samples from outpatients or hospitalized patients at the Hamamatsu University Hospital (Hamamatsu, Shizuoka, Japan), who requested clinical laboratory testing for TARC, mainly for the diagnosis of skin diseases, such as AD, and therapeutic monitoring (Table 1). Of the 452 patients whose TARC levels were measured from November 2020 to March 2021, those exceeding the measurement range, those with insufficient sample volume, and those with defective sample properties, such as hemolysis, were excluded; the remaining samples were randomly selected. In addition, for the healthy control group, 201 residual serum samples from medical checkups of health care workers and other staff at the Hamamatsu University Hospital were used. Blood was collected in INSEPAC II-D SMD750SQ tubes (Sekisui Medical Co., Ltd., Tokyo, Japan), and serum was prepared using the supernatant obtained after centrifugation at 2380× *g* for 10 min. All the samples were stored at −80 °C after collection or preparation and used for batch analysis. To confirm repeatability and interference effect, pooled anonymized serum samples were prepared and used. In addition, the limit of blank (LOB), the limit of detection (LOD), the limit of quantification (LOQ), and linearity, were determined using samples diluted with the TARC antigen and bovine-serum-albumin (BSA)-added buffer. Furthermore, the TARC value distribution in each disease and the concordance rate, sensitivity, and specificity of the two methods for AD were calculated for the abovementioned samples, and clinical evaluation was performed. The precision, accuracy, LOB, LOD, LOQ, linearity, interference, and commutability, which are indicators of analytical performance, were evaluated according to the CLSI protocols, EP05-A3, EP21, EP17-A2, EP06, EP7, and EP9c, respectively [23,24,25,26,27,28].

### 2.2. The Novel Latex Turbidimetric Immunoassay “Nanopia TARC”

The newly developed immunoassay, “Nanopia TARC” (Sekisui Medical Co., Ltd., Tokyo, Japan), is based on the LTIA method, which involves two types of latex compositions with different particle sizes. This homogeneous method is based on the principle of detection of agglutination caused by an antigen–antibody reaction between latex particles bound with a mouse monoclonal anti-human TARC and serum TARC, with the need for separation of the bound and free forms. Any general clinical chemistry analyzer can be used for the detection of agglutination (Appendix A). In this study, we used the Hitachi LABOSPECT008α (Hitachi High-Tech Co., Ltd., Tokyo, Japan), which is a general clinical chemistry analyzer. The measurement conditions were set according to the manufacturer’s specifications (Appendix A). Measurement time of Nanopia TARC is as fast as 10 min, similar to that for other measurements (e.g., creatinine) with a general clinical chemistry analyzer. The TARC antigen easily adsorbs to glass and probes and, therefore, an additional sample probe-washing step with HICARRYNON (citric acid monohydrate < 10%, oxyethylene = alkyl ether 5%; Hitachi High-Tech Co., Ltd., Tokyo, Japan) was set to prevent carryover in each specimen and ensure analytical precision.

### 2.3. “HISCL TARC” Based on the CLEIA Using HISCL^®^-5000

The reference measurement system for this study was “HISCL TARC” based on CLEIA using a two-step sandwich principle, with HISCL^®^-5000 as a dedicated device (Sysmex Corporation, Kobe, Japan) [29,30]. The biotin-labeled mouse monoclonal anti-human TARC specifically reacted with the serum TARC and then bound to streptavidin-bound magnetic particles. The unreacted solution was removed and washed repeatedly with the HISCL^®^-Washing Solution (for the separation of bound/free forms), and the luminescence intensity emanating from the reaction between the alkaline phosphatase-labeled anti-human TARC and its substrate, CDP-Star(R), was detected. The time required for HISCL TARC measurement was 17 min.

### 2.4. Analytical Precision and Accuracy

For determining the precision, repeatability was calculated as the coefficient of variation (CV) using the mean and standard deviation (SD) of 20 replicate measurements for three control samples and two pool serum samples. Intermediate precision was similarly calculated using two control samples measured thrice a day for a total of 10 d [19]. Accuracy was calculated by comparing with three reference materials targeted by the manufacturer’s (Sekisui Medical Co., Ltd., Tokyo, Japan) selected measurement procedures, and the total error (TE) was calculated as proposed by Westgard et al. [31]. 

### 2.5. LOB, LOD, and LOQ

For determining the LOB, LOD, and LOQ, 11 samples were prepared via serial dilution using the pooled serum and BSA-added buffer as a blank (theoretical values: 0, 95, 113, 127, 143, 159, 175, 191, 223, 254, and 318 pg/mL) and analyzed twice daily for 5 d to obtain a total of 110 measurements. According to CLSI EP17-A2, the LOD was calculated by combining the LOB with the SD of the estimated detectable measurement (1). Furthermore, the LOQ was used to determine the concentration at which the precision profile had a coefficient of variation (CV) of ≤20% [25].
LOD = LOB + 1.645 * × SD(1)
* where 1.645 is a constant indicating the 95th percentile for β = 0.05 in a normal distribution.

### 2.6. Linearity

For determination of linearity, 11 samples were prepared via serial dilution with the high-concentration TARC antigen and BSA-added buffer as a blank (theoretical values: 0, 2200, 4401, 6601, 8802, 11,002, 13,202, 15,403, 17,603, 19,804, and 22,004 pg/mL). These samples were measured in triplicate, and regression curves were evaluated using the least squares method and analysis of variance (ANOVA) to the theoretical values [26].

### 2.7. Interference

To examine the effects of hemolysis, chyle, bilirubin, and rheumatoid factor (RF) (TARC value: approximately 700, 1, and 800 pg/mL), 30 pooled serum samples were prepared using Interference Check A/RF Plus (Sysmex Corporation, Kobe, Japan) in accordance with the manufacturer’s instructions. These samples were measured in triplicate, and the relative difference (%) from the unspiked sample was calculated. The allowable relative difference was within ±10% [27].

### 2.8. Commutability and Clinical Evaluation

Method comparison studies were performed using 174 patient serum samples (Table 1). The values determined using Nanopia TARC were compared with those measured using HISCL TARC, employing the Passing–Bablok regression analysis [28].

In addition, from the distribution of TARC measurement values for each disease in Table 1, differences between the two methods were investigated for *n* > 10 disease groups. In particular, for AD, we calculated the concordance rate (%) of the two methods with respect to the cutoff values for children and adults (aged 6 years to <12 months: 1367 pg/mL; aged between 1 and <2 years: 998 pg/mL; aged between 2 and 15 years: 743 pg/mL; aged > 15 years: 450 pg/mL). Furthermore, the area under the curve (AUC), sensitivity/specificity, and cutoff values were calculated using the receiver operating characteristic (ROC) curve for adult-AD (*n* = 49) and healthy control groups, respectively, and differences between the two methods were investigated.

### 2.9. Statistical Analysis

All statistical analyses were performed using Microsoft Excel, Validation-Support/Excel Ver. 61 (the Japan Society of Clinical Chemistry, quality management expert committee: http://jscc-jp.gr.jp/?page_id=1145 (accessed on 1 September 2023)) and “EZR” (Easy R) based on statistics “R” (http://www.jichi.ac.jp/saitama-sct/SaitamaHP.files/statmedEN.html (accessed on 1 September 2023)). The 95% confidence interval was calculated via bootstrapping. The significance level for ANOVA used to determine linearity, Mann–Whitney *U* test, and difference between two AUCs was set at *p* < 0.001. For the Passing–Bablok regression used to determine commutability, a Pearson’s correlation coefficient (*r*) of 0.95 or higher and a slope of 0.9–1.1 were allowed.

## 3. Results

### 3.1. Precision and Accuracy 

The CV of the repeatability (CVr) was 3.8, 1.7, 1.0, 3.6, and 0.9%, respectively, in three control (mean; 618, 1821, 9830 pg/mL) and two pooled serum (mean: 823, 2069 pg/mL) samples. The CV of the intermediate precision (CVi) was 5.4 and 2.1% in two control samples, respectively (Table 2). In addition, the TE of the accuracy evaluation samples was 13.4, 3.7, and 1.9%, respectively (Table 3).

### 3.2. LOB, LOD, LOQ, Linearity, and Interference

The values of LOB, LOD, and LOQ were 13, 57, and 141 pg/mL, respectively (Figure 1). The linearity was accepted up to 19,804 pg/mL, where there was no statistically significant difference from the theoretical value (*p* = 0.080). Furthermore, the regression line also indicated very good linearity (Figure 2). Based on these results, the commutability study was performed by diluting samples to a concentration greater than 19,804 pg/mL. No interference was observed up to 500 mg/dL, 20 mg/dL, 2000 FTU, and 200 IU/mL for hemolytic hemoglobin, bilirubin, chyle, and rheumatoid factor, respectively, and the relative difference was below 10%. On the contrary, in the Interference Check RF Plus, the effect of rheumatoid factor increased in a dose-dependent manner (Figure 3a–e). 

### 3.3. Commutability and Clinical Evaluation

The correlation coefficient between Nanopia TARC and HISCL TARC determined using the Passing–Bablok regression analysis was 0.999, indicating a very good correlation. The slope of the regression line was 1.02 (95% confidence intervals (Cl): 1.01–1.04), which satisfies the criterion for statistical analysis (Figure 4a). The commutability at low to medium concentrations (less than 10,000 pg/mL) was almost the same (Figure 4b). 

There was no significant difference in the distribution of each disease between the two methods (Appendix A). The concordance rate with the JDA diagnostic criteria for AD [17] was as high as 92% (Table 4). The ability of Nanopia TARC to diagnose adult-AD assessed using the ROC analysis was indicated by an AUC of 0.932 (95% Cl: 0.887–0.976), sensitivity of 89.6%, and specificity of 85.1%, showing a statistically significant difference compared with HISCL TARC (Figure 5).

## 4. Discussion

Studies over the last 20 years have shown the clinical importance of TARC in many diseases, such as AD and cutaneous lymphoma. In Japan, it is used for the diagnosis of AD and is routinely used as a clinical laboratory test. In this study, we developed an LTIA method that could measure TARC levels rapidly (within 10 min) and easily. To the best of our knowledge, this is the first report showing equivalent analytical performance of a clinical laboratory test and a CLEIA method for TARC. In particular, owing to the advantage that this method can be applied in a general clinical chemistry analyzer, it can be used for diagnosing and monitoring via various brands of chemistry analyzers without the restrictions of a dedicated device.

Kataoka et al. [32] highlighted the significance of monitoring serum TARC levels during early intervention for severe infantile AD in determining the initial disease activity and for evaluating the treatment efficacy. Appropriate control of severe, early-onset infantile AD is important for improving the prognosis of eczema and for preventing food allergies. Additionally, TARC is a useful biomarker for improving patient adherence [32]. Moreover, the International Eczema Council conducted surveillance on the utilization of diagnostic biomarkers for AD [33]. It was noted that biomarker utilization for AD was only 29.55%, with IgE being the predominantly used biomarker. The percentage of people who responded affirmatively to the question of whether a blood test is useful for diagnosis and monitoring was very high, ranging from 59.09% to 76.74%. Furthermore, 52.9% of the respondents answered that they would want to have their TARC levels measured. Based on the above observations, Nanopia TARC may serve as a method for obtaining large amounts of data, such as in large-cohort studies, by establishing the TARC measurement system for clinical laboratory testing.

Conventionally, trace biocomponents have been measured using immunoassay systems, requiring ELISA plates and dedicated equipment. Because the range of absorbance analysis is from mmol/L to nmol/L for trace biocomponents, it is difficult for a general clinical chemistry analyzer to quantify them with sufficient sensitivity [34]. For procalcitonin, a trace biocomponent, Dipalo et al. and Yuan et al. [35,36] demonstrated an LOQ of 0.15–0.26 ng/mL (11.6–19.3 pmol/L) using the LTIA method. Nanopia TARC also had the same sensitivity as other LTIA methods and exhibited excellent linearity. To ensure precision, we utilized the sample wash function, which is a standard function in general clinical chemistry analyzers of several manufacturers. In the present evaluation, the low-level CVi was slightly above 5%; however, the biological variation of other cytokines/chemokines and TARC had a CVi of 14.15%. Both repeatability and intermediate precision satisfied the evaluation criteria. Recent studies have reported that TARC is present at low levels (below 100 pg/mL) in patients with severe COVID-19 [29,30]. Both Nanopia TARC and HISCL TARC are not effective in this measurement range.

With regard to linearity, the automated immunoassay reagent, HISCL TARC, has an upper limit of quantitation of 30,000 pg/mL. Although the performance range of this method is slightly narrow, it is applicable based on the diagnostic values and severity evaluation cut-off values, according to previous studies and guidelines on AD [10,11,12,13,14,15,16,17,18,19,20,21,22]. Moreover, cases with extremely high levels (10,000–30,000 pg/mL) of TARC have been reported to be correlated with the progression of cutaneous lymphomas, such as mycosis fungoides and *Sézary* syndrome [6,7]. In this study, three patients had mycosis fungoides, with TARC values exceeding 50,000 pg/mL. By diluting the samples and measuring the TARC levels, we confirmed the consistency of the results obtained with Nanopia TARC and HISCL TARC (Figure 4a). One of the limitations associated with the homogeneous method is the influence of interfering substances. As a mitigation measure, Nanopia TARC uses two detection wavelengths (main-/sub-) in the analyzer (Appendix A), which is suggested to decrease nonspecificity. In this study, the interference check, A/RF plus, which is used in many studies for the evaluation of analytical performance, was used. Although the results met the standard, caution is warranted with a sample having high levels of RF, exceeding 200 IU/mL. It is expected that the effects of self/heterophilic antibodies, such as a high-level RF, will be clarified in the process of measuring clinical specimens in the future.

Finally, the commutability between Nanopia TARC and HISCL TARC, which are comparative methods, fully satisfied the evaluation criteria in this study. The two methods are in concordance with the clinical diagnostic criteria for AD, indicating consistency with the findings reported to date. The clinical evaluation in this study also suggests that the ability to diagnose AD using ROC analysis had equal or better performance, although it would be necessary to increase the sample size in the future. Recent studies have provided evidence for an association between drug eruptions and TARC [37,38,39,40]. Drug eruption is a critical condition requiring immediate diagnosis and treatment. In a previous study, the clinical diagnostic value for drug eruption was set at 13,900 pg/mL [40]. This development, which can benefit from rapid reporting anywhere, may enhance the usefulness of TARC biomarkers for drug eruptions.

## 5. Limitations

This study had a limitation. The standardization and harmonization of the TARC value could not be achieved. It corresponds to Category 6 in the ISO 17511 classification [41]; therefore, the accuracy cannot be evaluated sufficiently. Accuracy was ≤15% TE for all samples, but a constant low-value bias (bias −56.0, −40.2, −96.6 pg/mL), independent of concentration, was observed. Even in previous studies, the commutability with ELISA reagents for TARC from R&D and Invitrogen and other fully automated immunoassay reagents was unclear. The clarification of the commutability between Nanopia TARC and HISCL will not only improve this bias but may also serve as an opportunity for future global standardization of the detection of TARC.

## 6. Conclusions

The TARC measurement system, “Nanopia TARC”, based on the LTIA method enabled rapid and easy measurement with a general clinical chemistry analyzer and was confirmed to have the same analytical performance as the existing CLEIA method as an AD biomarker. Thus, Nanopia TARC is a potential tool for establishing evidence of TARC as an AD biomarker.

## Figures and Tables

**Figure 1 diagnostics-13-02935-f001:**
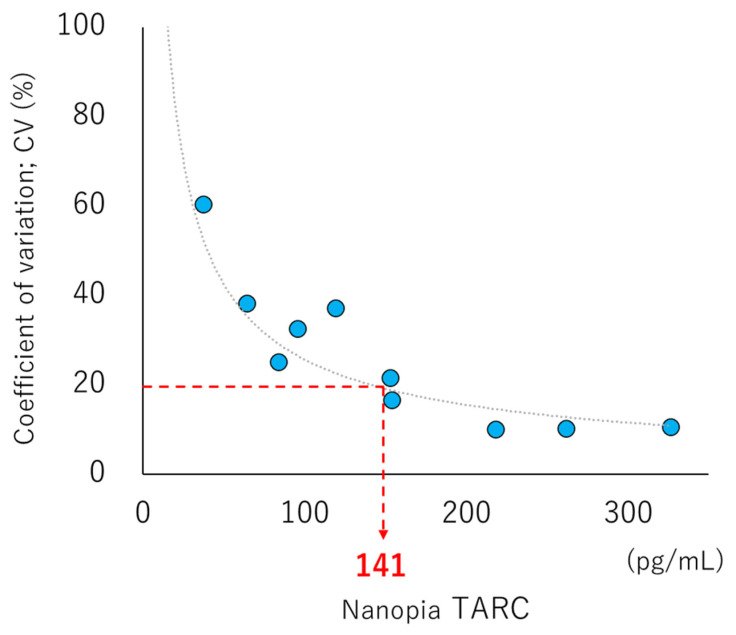
Limit of quantitation (LOQ) of Nanopia TARC based on the latex turbidimetric immunoassay method. Quantitative detectability is indicated by the total average (X-axis) of each evaluation sample (blue plot), excluding blanks, and its coefficient of variation (CV%; Y-axis) of repeated measurements. The dashed gray line indicates the fitted curve. The red dashed line indicates the threshold CV (20%) set for LOQ and the corresponding Nanopia TARC value.

**Figure 2 diagnostics-13-02935-f002:**
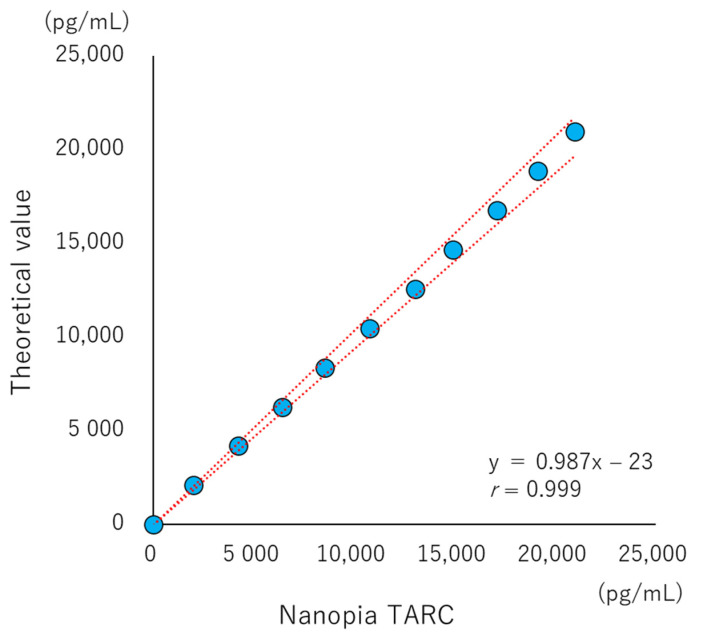
Linearity of the Nanopia TARC using the latex turbidimetric immunoassay method. The upper limit of the measurement range is indicated by the theoretical value (Y-axis) and the measured value (X-axis) of serially diluted evaluation sample (blue plot). The red dashed line indicates the 95% confidence interval of the regression line.

**Figure 3 diagnostics-13-02935-f003:**
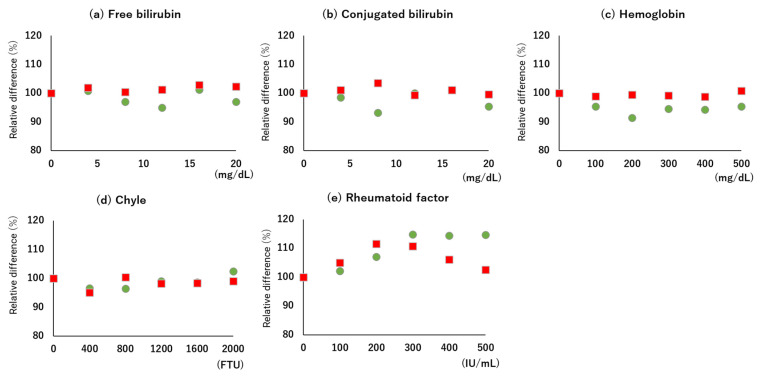
Interference of the Nanopia TARC using the latex turbidimetric immunoassay method. The effect of interference by (**a**) free bilirubin, (**b**) conjugated bilirubin, (**c**) hemolytic hemoglobin, (**d**) chyle, and (**e**) rheumatoid factor. The effect of concentration (X-axis) is shown as a relative value (Y-axis) wherein the relative value in the absence of interfering substance is set as 100%. Green circles indicate low (approximately 700 pg/mL) TARC values. Red squares indicate intermediate (approximately 1800 pg/mL) TARC values.

**Figure 4 diagnostics-13-02935-f004:**
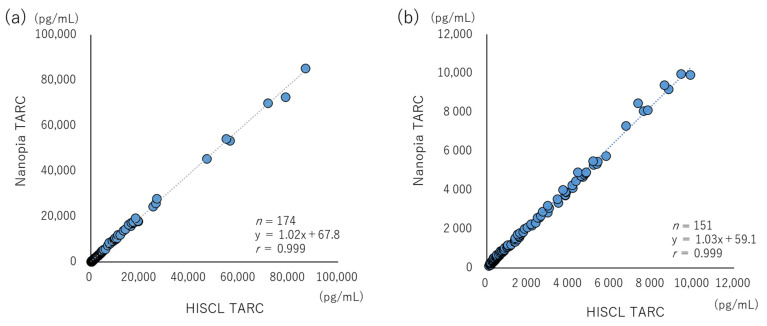
Commutability between Nanopia TARC (latex turbidimetric immunoassay) and HISCL TARC (chemiluminescent enzyme immunoassay). Scatterplot and regression analyses between Nanopia TARC (Y-axis) and HISCL TARC (X-axis) are presented. Blue plots are for patient samples and gray dashed lines indicate regression lines. (**a**) All samples, including those requiring a dilution assay (*n* = 174). (**b**) Samples with a measurement range of <10,000 pg/mL with intermediate to high levels of TARC (*n* = 151).

**Figure 5 diagnostics-13-02935-f005:**
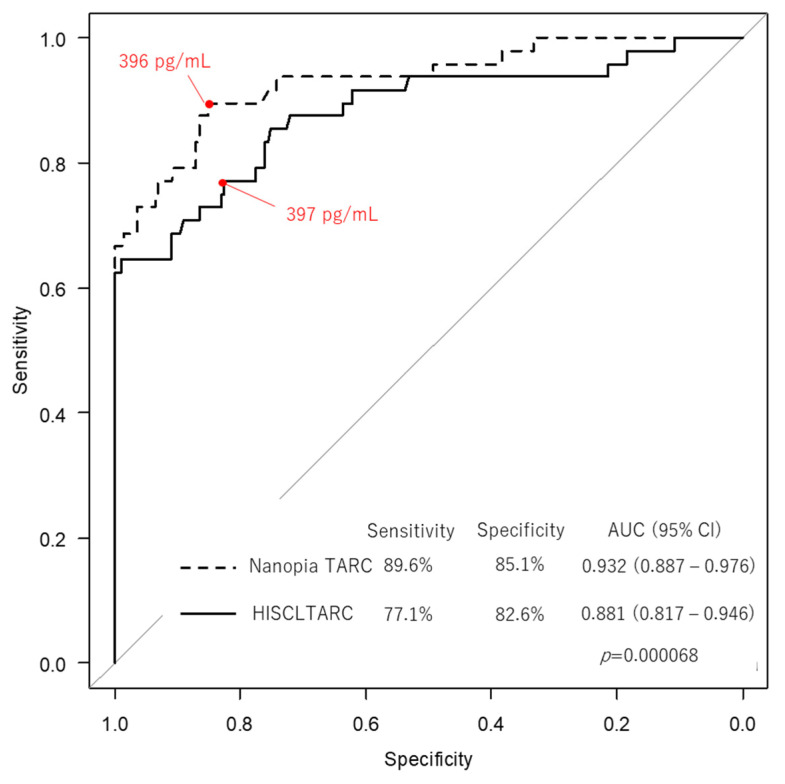
Comparison of the receiver operating characteristic curve between Nanopia TARC (latex turbidimetric immunoassay) and HISCL TARC (chemiluminescent enzyme immunoassay). The X- and Y-axes indicate specificity and sensitivity, respectively. Red indicates cut-off values for the two methods. The dashed line indicates the curve for Nanopia TARC and the black line indicates it for HISCL TARC. The *p*-value indicates the difference between the area under the curve (AUC) for the two methods.

**Table 1 diagnostics-13-02935-t001:** Characteristics of patients who requested thymus- and activation-regulated chemokine measurement.

Characteristics	Patients (*n* = 174)	Healthy Control (*n* = 201)
Age *: median (IQR), years	49 (48)	54 (9)
Sex		
Male	100	84
Female	52	117
Unknown	22	0
Disease	No. (%)	
Atopic dermatitis (AD)	60 (34)	
Non-AD eczema	4 (2)	
Non-AD prurigo	12 (7)	
Allergic rhinitis, asthma,	17 (10)	
and food allergy		
Mycosis fungoides	25 (14)	
*Sézary* syndrome	2 (1)	
Adult T-cell lymphoma	4 (2)	
Drug eruption	3 (2)	
Shingles (herpes zoster)	3 (2)	
Hives (urticaria)	2 (1)	
Bullous pemphigoid	3 (2)	
Psoriasis	2 (1)	
Other **	16 (9)	
unknown	22 (13)	

* Age at the time of blood sampling. ** Unclassifiable cutaneous symptoms. IQR: interquartile range.

**Table 2 diagnostics-13-02935-t002:** Precision of Nanopia TARC measurements.

*n* = 20 (pg/mL)	Control Samples			Pooled Sera	
	Low	Mid	High	Low	Mid
Mean	618	1821	9830	781	2045
SDr (pg/mL)	23	30	102	28	18
CVr (%)	3.8	1.7	1.0	3.6	0.9
SDi (pg/mL)	36	40	-	-	-
CVi (%)	5.4	2.1	-	-	-

SDr: standard deviation of repeatability. SDi: standard deviation of intermediate precision. CVr: coefficient of variation of repeatability. CVi: coefficient of variation of intermediate precision.

**Table 3 diagnostics-13-02935-t003:** Accuracy of Nanopia TARC measurements.

*n* = 5 (pg/mL)	Manufacture’s Reference Material		
	Low	Mid	High
Target value	693	1872	10,013
Bias (pg/mL)	−56	−40.2	−96.6
Bias (%)	−8.1	−2.1	−1.0
CV (%)	3.2	1.0	0.6
Total error; TE (%)	13.4	3.7	1.9

**Table 4 diagnostics-13-02935-t004:** Concordance with the diagnostic criteria for atopic dermatitis (AD).

Concordance Rate 92%	Nanopia TARC (Latex Turbidimetric Immunoassay)		
		<DC	≥DC
HISCL TARC (chemiluminescent enzyme immunoassay)	<DC *	16	5
	≥DC	0	38

* DC: Diagnostic criteria for AD (cut off). Aged from 6 to <12 months: <1367 pg/mL. Aged between 1 and <2 years: <998 pg/mL. Aged between 2 and 15 years: <743 pg/mL. Adults: <450 pg/mL.

## Data Availability

The data used and/or analyzed during the current study are available, only for sections of non-infringing personal information, from the corresponding author on a reasonable request.

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
