# Peer review of "Analytical Performance of a Novel Latex Turbidimetric Immunoassay, “Nanopia TARC”, for TARC/CCL17 Measurement: A Retrospective Observational Study"

_diagnostics, 2023, doi:10.3390/diagnostics13182935_

Round 1
Reviewer 1 Report
This manuscript reports that the new Nanopia method is a fast and reliable way to measure serum TARC levels. It seems that this an is innovative and useful method, but there are couple of questions and comments.
1) In Table 1, it is not clear “age, sex, and median”. What is age? Where are median values?
2) In Materials and Methods section, how were sera prepared? How were sera isolated?
3) Compared to the HISCL (CLEIA) method, what are sensitivity and specificity of Naopia method?
Author Response
Response to Reviewer 1’s Comments
We deeply appreciate your informative and kind comments, which have helped us to improve the quality of our manuscript. We have read the comments thoroughly and revised our manuscript accordingly.
Our revisions to the manuscript have been indicated using the “track changes” feature of MS Word and red text.
Comment: (→ Response)
- In Table 1, it is not clear “age, sex, and median”. What is age? Where are median values?
→Thank you for pointing this out. The composition of Table 1 have been changed significantly to clarify for age, median, IQR, sex, and disease classification. Based on the points mentioned in 3) below, we also have added information on the healthy control group (Line 107 - 110 in the revised manuscript).
- In Materials and Methods section, how were sera prepared? How were sera isolated?
→Thank you for pointing this out. We have made the indicated change and add.
(Line 110 - 112 in the revised manuscript)
- Compared to the HISCL (CLEIA) method, what are sensitivity and specificity of Naopia method?
→Thank you for pointing this out. The sensitivity and specificity of two methods were calculated, and Figure 5 of receiver operating characteristic analysis was have been added. Furthermore, we have added to Figure suppl 2, which shows the differences in the distribution of each disease shown in Table 1, and have made the indicated change and add to the methods, statistical analysis, results, and discussion.
(Lines 117 - 120, 194, 198 - 206, 211 - 215, 266, 273 - 278, and 358 - 360 in the revised manuscript)
Thank you for your consideration. Sincerely,
Keita Yamashita
Department of Laboratory Medicine, Hamamatsu University School of Medicine Hamamatsu, Japan
TEL: 81-53-435-2723
FAX: 81-53-435-2096
keitay@hama-med.ac.jp
-------------------------------------------------------
Reviewer 2 Report
1. Check the abbreviations throughout the manuscript and introduce the abbreviation when the full word appears the first time in the abstract and the remaining for the text and then use only the abbreviation (For example, ELISA, ANOVA, etc.,). Make a word abbreviated in the article that is repeated at least three times in the text, not all words to be abbreviated.
2. The introduction part appears less informative about atopic dermatitis, thus this section should be indicated as detailed to understand the manuscript in clear.
3. The authors may cite recent prevalence or incidence data atopic dermatitis and it should be at-least of 2022 or 2023.
4. The initial cited with reference in the text should be removed and should be in the author instruction of the journal (For example, “Kakimura T et al.”) and it should also be checked all over the manuscript.
5. The inclusion and exclusion criteria may be given detailed in methods and it is also advisable to include at the screening visit, how many total participants are included and out of them how many individuals are excluded for better understanding.
6. The limitation of the present research may be given along with conclusion or under separate heading for understanding the concepts clearly.
1. The English need improvement since there are some grammatical and syntax errors in the manuscript. For example,
· in line number 25, 79 and 80 the words “limit” may be as “the limit”;
· in line number , “were 0.9–3.8%” as “was 0.9–3.8%”;
· in line number 29, “were 2.1–5.4%” as “was 2.1–5.4%”;
· in line number 40, “the eosinophilic” as “eosinophilic”;
· in line number 51, “Japanese” as “the Japanese”;
· in line number 55, “the TARC” as “TARC”;
· in line number 133, “manufacturer's” as “the manufacturer's”;
· in line number 207 and 213, “indicated” as “is indicated”;
· in line number 221, “low” as “a low”;
· in line number 222, “intermediate” as “an intermediate”;
· in line number 243, “clinical” as “the clinical”;
· in line number 245, “diagnosis” as “the diagnosis”;
· in line number 255, “prognosis” as “the prognosis”;
· in line number 257, “a surveillance” as “surveillance”.
The grammar mistakes which are not mentioned here are also to be checked and corrected properly.
2. There are some typing mistakes as well, and authors are advised to carefully proof-read the text. For example,
· in line number 19 and 37, the words “activation regulated” may be as “activation-regulated”;
· in line number 41, “lymphoma ,” as “lymphoma,”.
The typos not mentioned here are also to be checked and corrected properly.
Author Response
Response to Reviewer 2’s Comments
We deeply appreciate your informative and kind comments, which have helped us to improve the quality of our manuscript. We have read the comments thoroughly and revised our manuscript accordingly.
Our revisions to the manuscript have been indicated using the “track changes” feature of MS Word and red text. Moreover, to entire the manuscript has been edited for language and readability, as suggested.
Comment: (→ Response)
- Check the abbreviations throughout the manuscript and introduce the abbreviation when the full word appears the first time in the abstract and the remaining for the text and then use only the abbreviation (For example, ELISA, ANOVA, ,). Make a word abbreviated in the article that is repeated at least three times in the text, not all words to be abbreviated.
→Thank you for pointing this out. We have checked the abbreviations to entire the manuscript and have corrected them as suggested (For example , abstract, lines 83, 87, 94, and 184 in the revised manuscript).
- The introduction part appears less informative about atopic dermatitis, thus this section should be indicated as detailed to understand the manuscript in clear.
→Thank you for pointing this out. We have made the indicated change and add, and the citation number has been changed accordingly.
(Line 50 - 63 in the revised manuscript)
- The authors may cite recent prevalence or incidence data atopic dermatitis and it should be at-least of 2022 or 2023.
→Thank you for pointing this out. We have made the indicated change and add, and the citation number has been changed accordingly.
(Line 64 - 75 in the revised manuscript)
- The initial cited with reference in the text should be removed and should be in the author instruction of the journal (For example, “Kakimura T et al.”) and it should also be checked all over the manuscript.
→Thank you for pointing this out. We have made the indicated change to entire manuscript for the relevant sections. (For example, Line 42, 165, 307, and 326 in the revised manuscript)
- The inclusion and exclusion criteria may be given detailed in methods and it is also advisable to include at the screening visit, how many total participants are included and out of them how many individuals are excluded for better understanding.
→Thank you for pointing this out. We have made the indicated change and add.
(Line 104 - 107 in the revised manuscript)
- The limitation of the present research may be given along with conclusion or under separate heading for understanding the concepts clearly.
→Thank you for pointing this out. We have made a separate heading for limitations of the study.
(Line 366 - 375 in the revised manuscript)
Comments on the Quality of English Language
- The English need improvement since there are some grammatical and syntax errors in the For example,
- in line number (of the revised manuscript) 25, 115 and 116 the words “limit” may be as “the limit”;
- in line number (of the revised manuscript) 28, “were 9–3.8%” as “was 0.9–3.8%”;
- in line number (of the revised manuscript) 29, “were 1–5.4%” as “was 2.1–5.4%”;
- in line number (of the revised manuscript) 40, “the eosinophilic” as “eosinophilic”;
- in line number (of the revised manuscript) 76, “Japanese” as “the Japanese”;
- in line number (of the revised manuscript) 79, “the TARC” as “TARC”;
- in line number (of the revised manuscript) 163, “manufacturer's” as “the manufacturer's”;
- in line number (of the revised manuscript) 250 and 256, “indicated” as “is indicated”;
- in line number (of the revised manuscript) 264, “low” as “a low”;
- in line number (of the revised manuscript) 265, “intermediate” as “an intermediate”;
- in line number (of the revised manuscript) 298, “clinical” as “the clinical”;
- in line number (of the revised manuscript) 299, “diagnosis” as “the diagnosis”;
- in line number (of the revised manuscript) 310, “prognosis” as “the prognosis”;
- in line number (of the revised manuscript) 313, “a surveillance” as “surveillance”.
The grammar mistakes which are not mentioned here are also to be checked and corrected properly.
- There are some typing mistakes as well, and authors are advised to carefully proof-read the For example,
- in line number 19 and 37, the words “activation regulated” may be as “activation-regulated”;
- in line number 41, “lymphoma ,” as “lymphoma,”.
The typos not mentioned here are also to be checked and corrected properly.
→Thank you for pointing this out. I have corrected all of the above points and edited the language of the entire manuscript.
Thank you for your consideration.
Sincerely,
Keita Yamashita
Department of Laboratory Medicine, Hamamatsu University School of Medicine Hamamatsu, Japan
TEL: 81-53-435-2723
FAX: 81-53-435-2096
keitay@hama-med.ac.jp
==============================================
Round 2
Reviewer 2 Report
1. There are some grammatical, alignments and typographical errors are noted in the manuscript and it should be thoroughly checked and corrected throughout the manuscript. For example,
· in line number 55, the words “a diverse” may be as “diverse”;
· in line number 66, “the childhood” as “childhood”;
· in line number 75, “the guidelines” as “guidelines”;
· in line number 80, “knowledge on” as “knowledge of”;
· in line number 202, “area” as “the area”;
· in line number 283, “level of” as “levels of”;
· in line number 327, “an excellent” as “excellent”.